# Carbene-activated stannylenes to access selective C(sp³)−H bond scission at the steric limit

Jennifer Klaucke[1], Navutheya Sinthathurai ®[1], Christopher Golz ®[2], Oliver P. E. Townrow ®[3] ✉ & Malte Fischer ®[1] ✉

The ubiquity of *N*-heterocyclic carbenes (NHCs) in diverse areas of chemical research typically arises from their potent stabilising capabilities and role as innocent spectators to stabilise otherwise non-bottleable compounds and complexes. This has, until now, been particularly true for NHC-stabilised stannylenes, with no exceptions reported thus far. Herein, we demonstrate that the combination of heteroleptic terphenyl-/amido-based stannylenes and tetra-alkyl substituted NHCs renders the corresponding NHC-ligated stannylenes highly reactive, yet isolable. In solution, this induces sterically controlled inter- and intramolecular C(sp³)−H bond scissions, resulting in the selective formation of stannylene metallocycles that depend on both the NHC source and the *meta*-terphenyl ligand coordinated to tin.

Since the first reports on the isolation of free singlet carbenes, stabilised by their incorporation in *N*-heterocycles[1], *N*-heterocyclic carbenes (NHCs) have become a fundamental cornerstone in molecular chemistry and beyond[2–11]. Whilst in transition metal chemistry they are known to enhance reactivity at the metal[1–12], in the field of main group chemistry they are typically used as an ambiphilic platform to stabilise low oxidation state and highly reactive compounds, acting as supporting ligands that do not directly participate in further reactivity[2–12]. In Group 14 chemistry, NHCs have been shown to allow for the isolation of the first dihalosilylenes, independently reported at the same time by Filippou, Roesky, Stalke and co-workers[13,14], which serve as soluble and readily available precursors of silicon(+II), with plenty of derivatives having been synthesized in subsequent years (**I**, Fig. 1)[2–12]. This is just one representative example, illustrating the impact of NHCs on Group 14 chemistry. Since then, for silicon and germanium, significant progress has been made in stabilising low-coordinate species and even allowed for the development of open-shell species[2–12].

On the contrary, silicon and germanium's heavier congener tin has received significantly less attention[2–12]. This can be attributed to poorer overlap between the NHC and the heavier tetrel elements, and consequently renders the formation of stable NHC–tin adducts more difficult compared to the lighter analogous[15]. Furthermore, the example of parent tetrylenes E(14)H₂ exhibiting decreased Lewis acidity when descending Group 14 can be referenced, making the formation of stable NHC−stannylene complexes more difficult compared to their lighter congeners[16].

Pioneering examples of NHC-stabilised stannylenes were reported by Kuhn et al. and Weidenbruch and co-workers, dating back to 1995[17,18]. They synthesized the dichloro- and diaryl-substituted stannylenes $Cl_2Sn(I^iPr_2Me_2)$ and $Tipp_2Sn(I^iPr_2Me_2)$, respectively ($I^iPr_2Me_2$ = 1,3-diisopropyl-4,5-dimethylimidazol-2-ylidene; Tipp = 2,4,6-$^iPr_3$-$C_6H_2$) (**II**, Fig. 1). This breakthrough opened the door to the isolation of numerous additional examples within this compound class, ranging from bis-hypermetallyl-substituted to transition metal-based derivatives[19–44].

All of these tin-containing examples share the characteristic that the NHC acts as an inert stabilising spectator ligand, not participating in any subsequent chemistry. As with any rule, there are exceptions. In this case, a comparatively small number of examples with other main group elements and transition metals have indeed shown follow-up reactivity, involving either formal C−H, C−N, or C−C bond activations; and ring opening, ring expansion, or protonation pathways[2–12].

[1]Institut für Anorganische Chemie, Georg-August-Universität Göttingen, Tammannstraße 4, D-37077 Göttingen, Germany. [2]Institut für Organische und Biomolekulare Chemie, Georg-August-Universität Göttingen, Tammannstraße 2, D-37077 Göttingen, Germany. [3]Institute of Nanotechnology, Karlsruher Institut für Technologie, Hermann-von-Helmholtz-Platz 1, D-76344 Eggenstein-Leopoldshafen, Germany. ✉e-mail: oliver.townrow@kit.edu; malte.fischer@uni-goettingen.de

Fig. 1 | **Notable literature examples and research scope. I**: Landmark NHC-stabilised dihalosilylenes; **II**: first examples of NHC-ligated stannylenes.

Here, we report the effects of NHC-ligation to heteroleptic stannylenes, which induce enhanced reactivity motifs, where the NHC is not merely an innocent spectator ligand. This led to formal inter- and intramolecular $C(sp^3)$–H activations at the backbone methyl groups of the chosen NHCs, leading to a diverse series of products. We show that these reactions are critically dependent on the vicinal steric hindrance of both the chosen NHC and the terphenyl ligands around the tin centre, leading to the selective formation of stannylene heterocycles and a series of macrocyclic multinuclear tin complexes.

## Results

### Synthesis and characterisation of 2

Expanding on our findings that the heteroleptic terphenyl-/amido-stannylenes of the type $^{Ar}TerSn\{N(SiMe_3)_2\}$ ($^{Ar}Ter = {}^{Mes}Ter = 2,6$-(2,4,6-$Me_2C_2H_6)_2C_6H_3$ (**1a**); $^{Ar}Ter = {}^{Dipp}Ter = 2,6$-(2,6-$^iPr_2C_6H_3)_2C_6H_3$ (**1b**); $^{Ar}Ter = {}^{Tipp}Ter = 2,6$-(2,4,6-$^iPr_3C_6H_2)_2C_6H_3$ (**1c**)) can formally activate $C(sp^3)$–H bonds through the elimination of the corresponding amine hexamethyldisilazane ($HN(SiMe_3)_2)^{45,46}$, we sought to investigate how the introduction of strong Lewis bases, such as NHCs, might impact the observed activation pattern and, ultimately, whether this could trigger the activation of the NHC. We hypothesized that NHC ligation renders the tin centre more electron-rich, thereby further weakening the tin–nitrogen bond while sterically repelling the trimethylsilyl groups. This, in turn, renders theses complexes more reactive towards C–H bond scission with the leaving group $HN(SiMe_3)_2$, thereby enhancing reactivity at the tin–nitrogen bond (Fig. 1, bottom). Hence, we initiated our study by treating **1a** with stoichiometric quantities of the NHC IMe$_4$ (1,3,4,5-tetramethylimidazol-2-ylidene). A rapidly advancing reaction was observed in benzene at room temperature, as evidenced by $^1H$ nuclear magnetic resonance (NMR) spectroscopy, indicating complete consumption of both starting materials (Fig. S10)[47]. This implied that the corresponding adduct $^{Mes}TerSn(IMe_4)\{N(SiMe_3)_2\}$ (**2a**) had been formed quantitatively (Fig. 2A). To our initial surprise, the characteristic signal of free $HN(SiMe_3)_2$ ($\delta^1H = 0.10$ ppm in $C_6D_6$) was observed with its intensity slowly increasing over time (cf. Fig. S16). Therefore, and to obtain **2a** in good isolated crystalline yields of 61%, the product was purified immediately by crystallisation of **2a** from Et$_2$O at -30 °C.

Yellow block-shaped crystals obtained via this method were suitable for single crystal X-ray diffraction (SCXRD) and verify the formation of the NHC-stabilised stannylene **2a** (Fig. 2C). **2a** crystallises in the monoclinic space group $P2_1/n$ with the tin atom being in a distorted trigonal pyramidal coordination environment ($\Sigma \sphericalangle Sn = 301.4°$). The dihedral angle between the planes defined by N1–Sn1–C8 and N2–C1–N3 is approximately 70°. The Sn–$C_{NHC}$ bond length of 2.3255(9) Å falls within the expected range for stannylene–NHC adducts. For comparison, it is slightly elongated relative to other terphenyl-substituted stannylenes cf. the tin(II)-monohydride $^{Mes}TerSn(IMe_4)H$ (2.2804(11) Å)[33], and slightly shorter than in the dichlorostannylene $Cl_2Sn(IPr)$ (2.341(8) Å; IPr = 1,3-bis(2,6-diisopropylphenyl)imidazol-2-ylidene)[22]. The Sn–$N_{amido}$ bond length (2.1990(8) Å) is elongated compared to **1a**, as well as the only other structurally characterised amido-substituted, NHC-stabilised stannylene IPrSn(Cl)N(H)Dipp (2.1142(16) Å)[26]. As another structural feature, the IMe$_4$ ligand and one mesityl group of the terphenyl moiety are aligned almost coplanar with a centroid–centroid distance of 3.48 Å, indicative of what is generally referred to as π–π-interactions (Fig. S37)[48,49].

In solution, this interplay seems to be maintained as shown by observed separate signals for both mesityl groups (Fig. S11). In the $^{119}Sn$ NMR spectrum of **2a** a singlet signal with a chemical shift of $\delta^{119}Sn = -15.7$ ppm is observed (Fig. S14), thus being significantly shifted towards higher field when compared to the starting material **1a** ($\delta^{119}Sn = 1192.3$ ppm; Fig. S1) and also when compared to other terphenyl- and amido-substituted NHC-stabilised stannylenes (IPrSn(Cl)N(H)Dipp ($\delta^{119}Sn = -93.2$ ppm); $^{Mes}TerSn(IMe_4)H$ ($\delta^{119}Sn = -349.4$ ppm))[26,33].

### Synthesis and characterisation of 3

Although the elimination of $HN(SiMe_3)_2$ already occurs at room temperature in solution, **2a** is stable for a prolonged time in the solid state under inert conditions (at least four weeks at –30 °C and one week at room temperature), allowing for the investigation of its reactivity. Complete consumption of **2a** accompanied by the release of $HN(SiMe_3)_2$ is realised by heating a solution of **2a** to 80 °C for 3 h, which is accompanied by the formation of a suspension from the initially clear solution (Fig. S15).

While this manuscript was in its final stages, the Inoue group also reported the isolation and characterisation of **2a**, observing that this compound "is unstable in solution and decomposes into an intractable

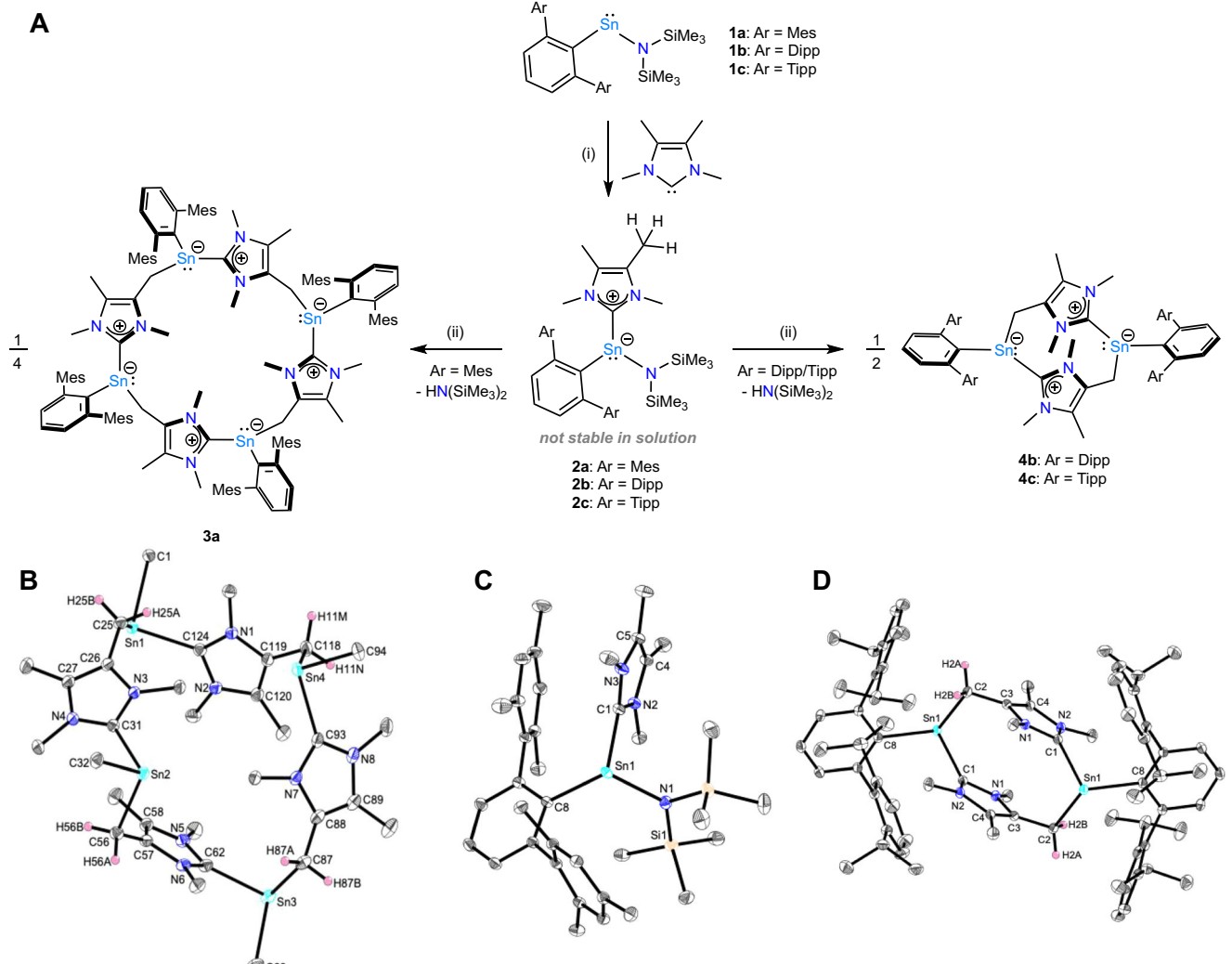

**Fig. 2 | Synthesis and structures of 2a-c, 3a, and 4b (polymorph a), and 4 c. A.** Reactivity of **1a-c**, towards IMe₄ to yield **2a-c**, the tetramer **3a**, and the dimers **4b,c**. Conditions (i) benzene, r.t.; (ii) benzene, r.t.–80 °C. **B–D**: Molecular structures of **2a**, **3a**, and **4a** determined by single crystal X-ray crystallography. Anisotropic displacement parameters are drawn at the 50% probability level. Hydrogen atoms have been omitted where necessary for clarity, and in the case of **3a** terphenyl substituents have also been omitted for clarity.

mixture of products after four hours at room temperature"[50]. As expected, the analytical data obtained by both Inoue and co-workers, as well as by us, are in excellent agreement, and a detailed discussion is presented here to support its categorisation within the context of this work. Notably, they further report on the use of **2a** as a hydrosilylation catalyst for aldehydes and ketones.

Upon slow evaporation of benzene solutions, SCXRD suitable colourless block-shaped crystals can be repeatedly obtained, revealing the formation of the tetranuclear, twenty-membered macrocycle **3a** as a result of consecutive C(sp³)–H-activations of the NHC backbone methyl groups (Fig. 2A, B). **3a** crystallises as a benzene solvate in the triclinic space group $P\bar{1}$ and can be regarded as self-assembled tetramer of a $^{Mes}$TerSnCH₂NHC building block. The shape of the macrocycle can be described as a "bowl", with two opposite "NHC-walls" facing inwards and outwards to varying degrees (Fig. S39). The distances between the tin atoms allow conclusions to be drawn about the approximate size of the "bowl" and are 9.33 Å (Sn1•••Sn3) and 6.15 Å (Sn2•••Sn4). Each tin atom is coordinated to three carbon atoms in a trigonal pyramidal coordination environment with C–Sn–C angles between 89° and 105°. All carbon–tin distances are in a narrow window of on average 2.26 Å

(Sn–CH₂), 2.27 Å (Sn–C$_{MesTer}$) and 2.28 Å (Sn–C$_{NHC}$) and, accordingly, slightly shortened when compared to the starting material **2a** (2.2809(9) Å (Sn–C$_{MesTer}$), 2.3255(9) Å (Sn–C$_{NHC}$)). Consequently, the NHC ligand in **2a** undergoes transformation into a unique formal hybrid NHC / mesoionic N-heterocyclic olefin (mNHO) ligand in **3a**, which serves to bridge the tin atoms[51,52]. mNHOs have recently emerged as exceptional ligands for main-group and transition-metal centres due to their outstanding donor properties and are typically generated through a methylation / deprotonation sequence, starting from abnormal NHCs[51].

Having found this unusual route towards macrocyclic main group systems based on a $^{Mes}$TerSnCH₂NHC building block generated upon amine release from an NHC-stabilised heteroleptic stannylene, we wanted to investigate how the choice of the ancillary ligands, namely the terphenyl substituent and the NHC, influence the macrocycle formation. Accordingly, $^{Dipp}$TerSn{N(SiMe₃)₂} (**1b**) and $^{Tipp}$TerSn{N(SiMe₃)₂} (**1c**) were reacted with IMe₄ in benzene or toluene at room temperature, leading to the immediate formation of the NHC-adducts $^{Dipp}$TerSn(IMe₄){N(SiMe₃)₂} (**2b**) and $^{Dipp}$TerSn(IMe₄){N(SiMe₃)₂} (**2c**), as verified by multinuclear NMR spectroscopy (δ$^{119}$Sn = -27.4 ppm (**2b**), -20.0 ppm (**2c**); Figs. S18 and S21) (Fig. 2A).

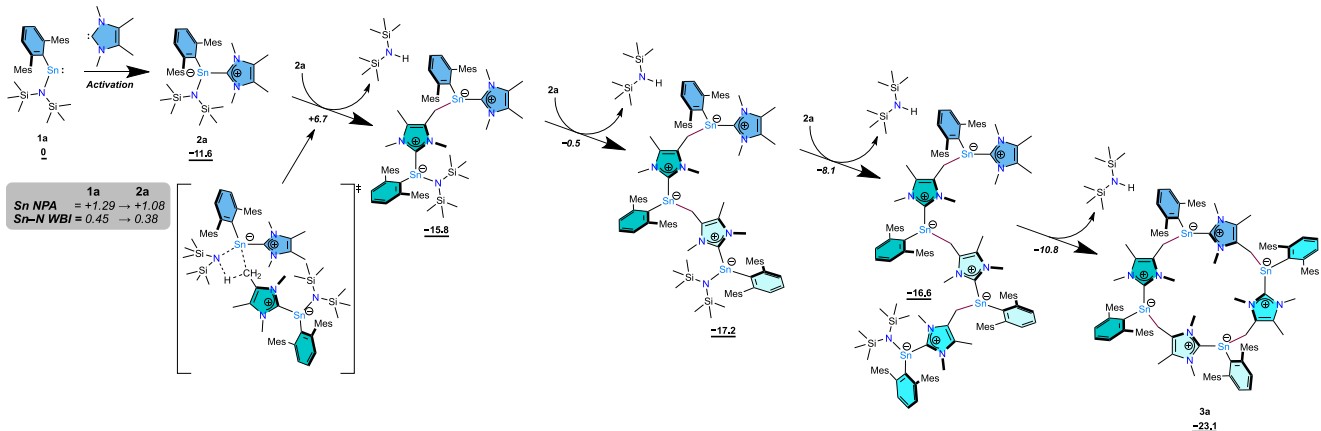

**Fig. 3 | Computed ring expansion mechanism.** DFT-calculated mechanism for the tin amide mediated C–H activation cascade forming **3a** at the BP86·D3BJ/Def2-TZVP (PCM = Benzene) level of theory. Gibbs free energies (ΔG₂₉₈, kcal•mol⁻¹) are given relative to the starting materials. Natural Population Analysis (NPA) charges and Wiberg Bond Indices (WBI) given for **1a** and **2a** were calculated using NBO-7.

## Synthesis and characterisation of 4

As observed for the analogous reaction with the ᴹᵉˢTer derivative, the formation of HN(SiMe₃)₂ already starts straight away and after **1b,c** were consumed completely, slow evaporation of benzene solutions led to the formation of colourless and yellow crystals, respectively, suitable for SCXRD, verifying the formation of the ten-membered macrocyclic dimers **4b** and **4c** of the ᴬʳTerSnCH₂NHC building block (Fig. 2D and Fig. S30). In analogy to **3a**, **4b,c** both crystallise in in the triclinic space group P1̄ as benzene solvates with the tin centres being in trigonal pyramidal coordination environments. For **4b**, crystals suitable for SCXRD were also successfully obtained without the co-crystallisation of any additional solvent (Fig. S24). Since the solvent-free crystal structure exhibits slight disorder, the structural parameters of the solvent-containing structure are discussed for clarity (both modifications show very good agreement in their structural parameters)[47]. The tin–carbon distances are further apart when compared to **3a**. The shortest bond length is observed for Sn–C_DippTer/TippTer (2.2420(9) Å (**4b**), 2.2444(11) Å (**4c**)) and the longest bond length is either observed for Sn–CH₂ (2.3111(10) Å) in case of **4b** or for Sn–C_NHC (2.3031(15) Å) in case of **4c**. The Sn•••Sn distances are 6.45 Å (**4b**) and 6.46 Å (**4c**), respectively, and as a structural feature, the two five-membered imidazolium moieties are not aligned coplanar but show a rather short distance to each other of approximately 3.40 Å (**4b**) and 3.47 Å (**4c**) by considering the centroids of the five-membered N-heterocycles.

It is worth mentioning that although **3a** and **4b,c** can be reproducibly obtained as single crystalline materials, sufficiently clean NMR spectra for the characterisation of both **3a** and **4b,c** in solution could not be obtained due to their poor solubility in various organic solvents (aliphatic and aromatic hydrocarbons, ethers etc.), especially when the crystalline materials were dried under vacuum. This is likely due to removal of the lattice benzene solvent observed during SCXRD upon drying *in vacuo*. Information on the bulk purity via elemental combustion analysis repeatedly led to good agreement of the theoretical and measured H and N values, with the C values consistently being too low[47].

Notably, **4c** demonstrated slightly increased solubility in organic solvents compared to all other herein reported macrocycles, allowing for the detection of its M⁺ signal by LIFDI mass spectrometry.

Interestingly, when **1a** was reacted with IPr – an NHC lacking a backbone methyl group – at temperatures up to 100 °C, only the tin starting material decomposed, while the NHC remained unaffected.

To obtain further insight into the unique formation of **3a** and **4b,c**, and bonding situation therein, quantum chemical calculations were performed at the BP86·D3BJ/Def2-TZVP (PCM = benzene) level of theory, which includes corrections for both solvation and dispersion effects[53–58]. On barrierless coordination of IMe₄, adducts **2a** and **2b** were found to form with a ΔG₂₉₈ of −11.6 and −10.7 kcal•mol⁻¹ respectively. Inspection of the Kohn-Sham HOMO indicates that the Sn(II) centre retains its lone pair character, owing to either zwitterionic or donor-acceptor natures of the C_NHC⁻–Sn interactions. This is comparable to our previously reported complexes ᴹᵉˢTerSn(IMe₄)CH₂PNR (R = alkyl, aryl)[46]. Natural Bond Orbital (NBO) analysis confirms this[59], providing a two electron NBO for the Sn lone pair. Like **1a**, a nitrogen-based lone pair also contributes to the HOMO which allows for the amide to also take part in nucleophilic processes. The virtual orbitals, on the other hand, are different: **1a** has a lower-lying LUMO (HOMO−LUMO gap = 2.344 eV), which exhibits p-orbital character at the tin centre, allowing for its donor-acceptor type reactivity. In **2a**, the NHC has extinguished this orbital, leading to a larger HOMO–LUMO gap (2.542 eV) and a terphenyl-centred delocalised LUMO. Meanwhile, the LUMO + 1 is localised on the NHC in its archetypal pπ form.

We have previously demonstrated the reactivity of the Sn–N moiety in **1** towards C–H bonds, resulting in the production of bis(-trimethylsilyl)amine and a tin alkyl complex, albeit with mildly activated C–H bonds[45,46]. Compounds **2**, on the other hand, are able to activate aryl-substituted methyl groups, which are typically more difficult to activate. Natural Population Analysis (NPA) found a significant decrease in NPA charge (1.29 to 1.08) at the tin centre, showing higher electron density at the metal (*i.e.* increased basicity). This is reflected in the Wiberg bond index for the Sn–N interaction, which decreases from 0.45 to 0.38, indicating a reduction in bond strength and thus allowing for the more facile activation across the bond; and supports the observed elongation of the bond measured by SCXRD, *vide supra*. A model mechanism for the formation of **3a** from **1a** was calculated using Kohn-Sham Density Functional Theory (DFT), finding an overall Gibbs free energy, ΔG₂₉₈, of −23.1 kcal•mol⁻¹. This cascade reaction consists of four consecutive C–H activation steps with decreasing activation energy, ΔG‡₂₉₈, between +18.2 and +5.7 kcal•mol⁻¹ (Fig. 3), which explains why the reaction is facile at room temperature and low temperatures are required to isolate **2a** without further activation.

The unobserved cyclic dimer, isostructural to **4b**, was found to be thermodynamically less favourable than the final tetrameric product with a ΔG₂₉₈ of −15.3 kcal•mol⁻¹[47]. On the contrary, the difference in energy between **4b** and the hypothetical tetramer was found to be less than 1 kcal•mol⁻¹, owing to its higher steric demand. To quantify how NHC coordination induces the observed increased reactivity, a

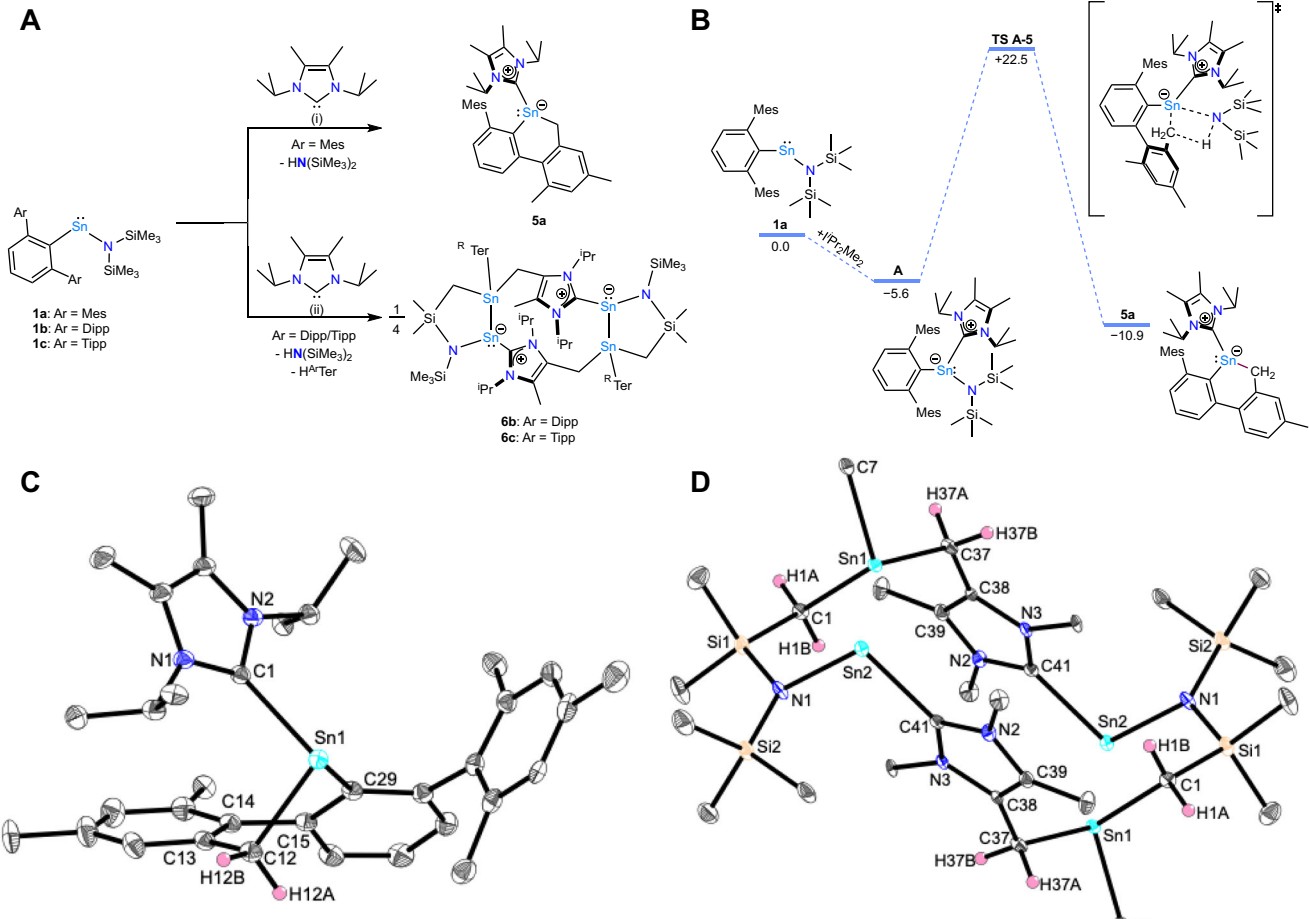

**Fig. 4 | Synthesis and structures of 5a, and 6b,c. A.** Reactivity of $^{Mes}$TerSn{N(SiMe₃)₂} (**1a**), $^{Dipp}$TerSn{N(SiMe₃)₂} (**1b**) and $^{Tipp}$TerSn{N(SiMe₃)₂} (**1c**) towards I$^i$Pr₂Me₂ to yield the C–H activation product **5a**, and doubly C–H activation products **6b,c**. Conditions (i) benzene, r.t.–60 °C; (ii) benzene, r.t.–90 °C. **B:** DFT-Calculated mechanism for the tuck-in formation of **5a** from **1a** and I$^i$Pr₂Me₂ at the BP86-D3BJ/Def2-TZVP (PCM = Benzene) level of theory. Gibbs free energies ($\Delta G_{298}$, kcal•mol$^{-1}$) are given relative to the starting materials. **C,D** Molecular structures of **5a** and **6b** determined by single crystal X-ray crystallography. Anisotropic displacement parameters are drawn at the 50% probability level. Hydrogen atoms have been omitted where necessary for clarity, and in the case of **6b** terphenyl substituents have also been omitted for clarity.

hypothetical reaction of **1a** and **2a** was modelled by DFT, where **1a** activates the C–H bond of the NHC in **2a**, analogous of the first C(sp³)–H activation step shown in Fig. 3. This was found to proceed through a much higher energy transition state ($\Delta G^{\ddagger}_{298} = 37.4$ kcal•mol$^{-1}$), roughly double the height of the barrier when two equivalents of **2a** react, *vide supra*, to an endergonic product ($\Delta G_{298} = 4.4$ kcal•mol$^{-1}$) (Fig. S55).

### Synthesis and characterisation of 5

After observing and analysing the impact of the different terphenyl ligands on the reactivity of the respective heteroleptic stannylenes **1a-c** towards IMe₄, the reactivity of **1a-c** towards the slightly larger NHC I$^i$Pr₂Me₂ (1,3-diisopropyl-4,5-dimethylimidazol-2-ylidene) was investigated. Upon equimolar addition of I$^i$Pr₂Me₂ to a solution of **1a** in benzene at room temperature, no reaction between the two substrates can be observed by NMR spectroscopy. However, heating the reaction mixture to 60 °C leads to clean conversion to a single species, accompanied by the release of HN(SiMe₃)₂. By contrast to the formation of the hitherto reported macrocycles **3a** and **4b,c**, the newly obtained compound exhibits good solubility in both aliphatic and aromatic hydrocarbons, enabling its characterization through multinuclear NMR spectroscopy. Additionally, crystals suitable for SCXRD were obtained from concentrated solutions in *n*-hexane at -30 °C, revealing that, unlike the smaller carbene, IMe₄, I$^i$Pr₂Me₂ induced an

intramolecular C(sp³)–H bond scission at the benzylic position of the terphenyl ligand, yielding the carbene-stabilised stannylene heterocycle **5a** (Fig. 4).

The monomeric NHC-ligated stannylene **5a** crystallises in the triclinic space group P$\bar{1}$. The tin atom is in a distorted pyramidal coordination environment (C1–Sn1–C12 98.96(7)°, C1–Sn1–C29 99.26(7)°, C12–Sn1–C29 80.83(7)°). The Sn–C$_{NHC}$ bond length of 2.316(2) Å is in good agreement to the other herein reported monomeric stannylene **2a** (2.3255(9) Å). The tin atom is embedded in a six-membered all-carbon heterocycle, thus representing an NHC-stabilised tin-carbon heterocycle. In this context the Sn–C(sp²) and Sn–C(sp³) bond lengths are almost identical (2.2470(19) Å and 2.2474(18) Å, respectively) and are exceeding the single bond covalent radii by approximately 0.10 Å ($\Sigma_{cov}$(Sn–C) = 2.15 Å$^{60,61}$). **5a** exhibits a $^{119}$Sn NMR chemical shift of $\delta^{119}$Sn = –164.4 ppm which is in good agreement with literature examples, *e.g.* the NHC-stabilised diaryl substituted stannylene Tipp₂Sn(IMe₄) ($\delta^{119}$Sn{$^1$H} = -160.7 ppm)$^{31}$.

The intramolecular C(sp³)–H activation found in the formation of **5a** was investigated using DFT (Fig. 4B). This showed that the initial coordination of the larger carbene ligand was stabilising by just $\Delta G_{298} = -5.6$ kcal•mol$^{-1}$, half the thermodynamic impact than its smaller analogue. This produced optimised intermediate is sterically restrained, with an even weaker Sn–N bond (Wiberg bond index = 0.34) and in addition, one of the methyl groups from the terphenyl

moiety in the vicinity of the activated $N_{amido}$. Abstraction of the proton then proceeds with a $\Delta G^{\ddagger}_{298}$ of +22.5 kcal·mol$^{-1}$, in line with requirement to heat the reaction, producing the amine and heterocycle **5a** with an $\Delta G_{298}$ of −10.9 kcal·mol$^{-1}$. Since stannylenes have been shown to be capable of oxidative addition[62], we also explored the two-electron redox pathway computationally, where oxidative addition of the C−H bond results in a Sn(IV) alkyl-hydride intermediate. This was found to be energetically much less favourable than the reported redox-neutral pathway ($\Delta G^{\ddagger}_{298,OxAdd}$ = +41.4 kcal·mol$^{-1}$), producing the Sn(IV) species with an $\Delta G_{298}$ of +19.5 kcal·mol$^{-1}$.

### Synthesis and characterisation of 6

Finally, we investigated the reactions of $^{Ar}$TerSn{N(SiMe$_3$)$_2$} (Ar = Dipp (**1b**), Ar = Tipp (**1c**)) with I$^i$Pr$_2$Me$_2$, the mixtures with the largest vicinal steric hindrance around the metal (Fig. 4). Upon mixing **1b,c** and the respective NHC in C$_6$D$_6$ at room temperature, no reactions could be observed within 16 hours. Reactivity was however enabled by heating to reflux, as verified by the detection of small amounts of HN(SiMe$_3$)$_2$ by $^1$H NMR spectroscopy (*cf.* Fig. S34). Further heating of the reaction mixtures caused the solutions to slowly turn cloudy, and two doublets in close proximity to each other ($\delta^1$H = 1.11-1.15 ppm) in addition to a new heptet signal ($\delta^1$H = 2.90 ppm) were observed in the $^1$H NMR spectrum, corresponding to free $^{Ar}$TerH (Figs. S26 and S27)[63,64]. **1b,c** were completely consumed after several days of heating. Notably, significant amounts of I$^i$Pr$_2$Me$_2$ remained unreacted when stoichiometric amounts were used in initial attempts. Gratifyingly, these reaction conditions repeatedly led to the precipitation of yellow crystals suitable for SCXRD when starting from **1b,c**, which could be separated by filtration. The bulk purity of these materials was confirmed by elemental combustion analysis of the corresponding yellow powders after drying under vacuum.

The newly obtained compound derived from **1b** crystallises in two different crystal habits (block-shaped and needle-shaped crystals), depending on the varying equivalents of co-crystallized benzene molecules. Both forms are highly sensitive, decomposing within minutes even when coated in NVH oil. Due to nearly identical structural data, the parameters of the needle-shaped crystals are discussed because of the better quality of the data set. The SCXRD results perfectly match the observations made while monitoring the reaction by NMR spectroscopy and demonstrate the formation of the tetranuclear tin macrocycle **6b** (Fig. 4D). Both polymorphs of **6b** crystallise in in the triclinic space group $P\bar{1}$. The release of both HN(SiMe$_3$)$_2$ and $^{Dipp}$TerH, along with the remaining I$^i$Pr$_2$Me$_2$ observed during reaction monitoring of the first attempt (*vide supra*), is confirmed by the connectivity of the macrocycle. The flanking five-membered dimetallacycles in **6b** adopt distorted tetrahedral geometries, as evidenced by the $\tau_4$ and $\tau_4'$ geometry indices of 0.79 and 0.72, respectively, owing to the inherent strain of the heterocycle[65,66]. This distortion is further indicated by the small Sn1−Sn2−N1 and Sn2−Sn1−C1 angles of 88.30(12)° and 92.59(11)°. The three-coordinate tin atom Sn1 is in a trigonal pyramidal coordination environment (N1−Sn2−Sn1 88.30(12)°, Sn2−Sn1−C1 92.59(14)°, N1−Sn2−C41 98.18(19)°). The Sn1−Sn2 bond length of 2.9055(9) Å exceeds the sum of covalent radii by approximately 0.10 Å ($\Sigma_{cov}$(Sn−Sn) = 2.80 Å[60,61]) and is also elongated compared to compounds featuring the same structure of the discussed five-membered ring (2.737(2) Å[67] and 2.870(1) Å[68]). While these literature-known examples formally feature Sn(III)−Sn(III) and Sn(III)−Sn(II) moieties, **6b** features a formal Sn(III)−Sn(I) bond. A slight elongation is also observed for the Sn−C$_{NHC}$ bond length of 2.335(6) Å in **6b**, compared to the other tin macrocycles reported herein (*cf.* 2.2988(9) Å in **4a**), likely due to the inherently different bonding situation at Sn2. The CH$_2$NHC units bridge the two five-membered ring systems, completing the tricyclic structure with a formal 12-membered central ring system. The structural data of the $^{Tipp}$Ter-substituted

derivative **6c** are, as expected, in very good agreement to those of **6b** (Fig. S36).

To further understand both the formation and bonding situation of this unique tricyclic compound, quantum chemical investigations were conducted. The most sterically restricting combination, **1b** and I$^i$Pr$_2$Me$_2$, is close to thermoneutral ($\Delta G_{298}$ = +0.4 kcal·mol$^{-1}$) in the formation of the adduct ($^{Dipp}$TerSn(I$^i$Pr$_2$Me$_2$){N(SiMe$_3$)$_2$}). Whilst a mechanism to the formation of the unique product **6b** could not be elucidated, the loss in adduct favourability can be rationalised by the activation of other bonds (such as the loss of the terphenyl moiety). The NHC-induced ligand loss in mono(aryl)tin compounds is known in the literature to stabilise via Sn−Sn bond formation, towards cluster assembly[30]. From a structure and bonding perspective, **6b** is dimeric, composed of two identical components, each containing two bonded Sn centres with very different bonding situations. The most interesting aspect is the Sn−Sn bond itself. NBO analysis describes two inequivalent Sn centres, Sn1 and Sn2: Sn1 is tetracoordinate with an NPA charge of +1.09 (similar to compounds **2**), while Sn2 is tricoordinate with an NPA charge almost half of that at +0.64, indicating an extremely electron rich centre. Sn2 also possesses a lone pair of electrons primarily with $s$ character (1.93 e$^-$, $s^{0.89}p^{0.11}$), in line with **2**. The Sn−Sn bond is characterised by two NBOs totalling 1.92 electrons, both polarised towards the tetracoordinate Sn centre Sn1 (64.3%). In this interaction, Sn1 uses a hybrid orbital of $s^{0.35}p^{0.65}$ composition, whereas Sn2 uses primarily p character ($s^{0.04}p^{0.96}$). As observed in compounds **2**, the NHC substituent is in an imidazolium resonance form, indicating a zwitterionic relationship with Sn2. Based on their NPA charges and connectivity, one could assign the oxidation states of these tin centres as Sn(III) and Sn(I), which can be compared to the structure of the literature-known stannyl-stannylene [[1,2-C$_6$H$_4${CHP(BH$_3$)Cy$_2$}$_2$]Sn]$_2$[69].

## Discussion

In summary, a synthetic protocol was developed to access a range of well-defined tin macrocycles and a cyclic NHC-stabilised stannylene. The outcomes of the reactions were highly dependent on both the NHC employed and the substitution pattern of the kinetically stabilising terphenyl ligands in the heteroleptic stannylenes used. This revealed a unique non-innocent behaviour of NHCs within the coordination sphere of stannylenes and their transformation into hybrid NHC / mNHO ligands, which bridge the tin centres, emerges as a predominantly observed motif. We are currently envisioning and working on the use of these macrocyclic systems in template-controlled sensing and capturing of appropriate building blocks.

## Methods

Detailed descriptions of experimental, spectroscopic, crystallographic and quantum chemical methods and results are given in the *Supplementary Information*. The authors have cited additional references in the *Supplementary Information*[70–79].

### General

All manipulations of air- and moisture-sensitive materials were carried out using standard Schlenk-line and glovebox techniques (MBraun glovebox with oxygen and water concentrations below 0.1 ppm as monitored by an O$_2$/H$_2$O Combi-Analyzer) under an inert atmosphere of argon.

### NMR spectroscopy

NMR spectra were measured in benzene-$d_6$ (C$_6$D$_6$) or toluene-$d_8$ (C$_7$D$_8$) (dried over CaH$_2$, distilled by trap-to-trap transfer in vacuo, degassed by three freeze-pump-thaw cycles and transferred to the glovebox). NMR samples were prepared under argon in NMR tubes with J. Young Teflon valves. NMR spectra were measured on Bruker Avance 400 MHz, 500 MHz and 600 MHz spectrometers. $^1$H and $^{13}$C NMR

spectra were referenced internally to residual protio-solvent ($^1$H) or solvent ($^{13}$C) resonances ($C_6D_6$: $d_H$ = 7.16 ppm; $d_C$ = 128.06 ppm; $C_7D_8$: 2.08 ppm; $d_C$ = 20.43 ppm). $^{119}$Sn NMR spectra were referenced with respect to $SnMe_4$.

## Mass spectrometry

LIFDI-(JEOL AccuTOF JMS-T100GCV; inert conditions) and ESI- (Bruker Daltronik micro TOF) MS were measured by the Zentrale Massenabteilung (Fakultät für Chemie, Georg-August-Universität Göttingen).

## Elemental analysis

Elemental analyses were obtained from the Analytische Labor (Georg-August-Universität Göttingen) using an Elementar Vario EL 3 analyzer.

## X-ray crystallography

Suitable crystals were selected and mounted on a MiTeGen micromount with NVH oil. Single crystal X-ray data were collected from shock-cooled single crystals at 100.00 K on a Bruker D8 VENTURE diffractometer equipped with an Oxford Cryostream 800 low temperature device. The used radiation sources are Incoatec I$\mu$S 2.0 or 3.0 microfocus sealed X-ray tubes using mirror optics as monochromators using Mo$K\alpha$ radiation ($\lambda$ = 0.71073 Å) and Bruker PHOTON III detectors.

## Computational methods

Geometry optimisations, frequency calculations and PCM solvent corrections were run with Gaussian 16 Revision A.03 using the BP86 functional. For geometry optimisations, all atoms were described with def2-SVP basis sets of Ahlrichs and Weigand. Single point energy calculations were performed on the optimised geometries, at the BP86/def2-TZVP level of theory. Stationary points were fully characterised using analytical frequency calculations as either minima (all positive eigenvalues) or transition states (one negative eigenvalue). IRC calculations and subsequent geometry optimisations were used to confirm the minima linked by the transition states. Energies reported in the text are based on the gas-phase free energies and incorporate a correction for dispersion effects using Grimme's D3 parameter set with Becke-Johnson dampening (*i.e.* BP86-D3BJ) as well as solvation (PCM approach) in benzene. Energies are given in atomic units (a.u.) unless otherwise stated. Natural Bond Orbital (NBO) and Natural Localised Molecular Orbital (NLMO) analysis was performed using NBO-7 using single point calculations performed at the BP86/def2-SVP or BP86/def2-TZVP level of theory.

## Online content

## Data availability

All data supporting this study are available within the paper and the *Supplementary Information*. Upon request, all data is available from the corresponding authors. Cartesian coordinates of the optimised geometries are available with this paper. Crystallographic data for the new structures reported in this article have been deposited at the Cambridge Crystallographic Data Centre, under depositions numbers CCDC 2386607 (**2a**), 2386608 (**3a**), 2386609 (**4b** (polymorph a)), 2411687 (**4b**) (polymorph b), 2386610 (**4c**), 2386611 (**5a**), 2386613 (**6b** (polymorph a)), 2386612 (**6b** (polymorph b)), and 2386614 (**6c**). Copies of the data can be obtained free of charge via https://www.ccdc.cam.ac.uk/structures/. Source data are provided with this paper.

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

## Acknowledgements

Financial support (VCI Liebig fellowships for M. F. and O. P. E. T.) is gratefully acknowledged. M. F. further wishes to acknowledge the Georg-August-Universität Göttingen for financial support and Prof. Inke Siewert for her continuous support and guidance. We also thank the NMR, MS and EA services at the Faculty of Chemistry (Georg-August-Universität Göttingen) for technical assistance. Support from the DFG (INST 186/1237-1 and INST 186/1324-1) is also gratefully acknowledged. The authors acknowledge support by the state of Baden-Württemberg through bwHPC and the DFG (INST 40/575-1 FUGG) for access to the JUSTUS 2 high performance computing cluster. We acknowledge support by the Open Access Publication Funds/tranformative agreements of the Göttingen University.

## Author contributions

J.K., N.S. and M.F. carried out the experimental work. C.G. performed the X-ray crystallographic work and M.F. and C.G. performed the crystallographic data analyses. O.P.E.T. carried out the quantum chemical calculations and wrote the initial draft of the computational part of the manuscript. M.F. was responsible for the conceptualisation, supervision of the experimental investigations, and wrote the initial draft of the manuscript. All authors contributed to the finalisation of the manuscript and agreed to the submitted content.

## Funding

## Competing interests

The authors declare no competing interests.
