## [Transparent Peer Review file · Nature Communications]

Carbene-activated stannylenes to access selective C(sp³)–H bond scission at the steric limit

Corresponding Author: Professor Malte Fischer

Version 0:

Reviewer comments:

Reviewer #1

(Remarks to the Author)

“Carbene-destabilised stannylenes: Accessing selective C(sp³)–H bond scission at the steric limit” article by Townrow, Fischer, et al. reports on the inter- and intramolecular activation of C-H bonds by a low oxidation state Sn center. The activation is induced upon the reaction of Sn(II) with NHCs. Overall, the work is of interest to organometallic and main-group chemists. Below are several points that the authors should address before the publication:

1. The principle reported here is conceptually similar to that reported already. The addition of a Lewis base facilitates the reactivity of the Sn(II) center that is accompanied by the release of HN(SiMe₃)₂ (Ref. 43,44). The authors should make the point of novelty more clearly.
2. One of my biggest concerns is about the presentation. I understand the authors try to make a catchy title, however in my opinion it really hurts the chemistry carried out here. Referring to the NHC-stannylene complexes as “carbene-destabilized” is very odd. I completely disagree with this type of statement. The complexation may make the Sn center more nucleophilic, and more reactive; it may induce reactivity of which the “naked” Sn complex is not capable. The introduction of bulkier substituent makes the Sn center more prone to reductive elimination – this is a known phenomenon. Steric effects from bulky ligands can increase the rate of reductive elimination by creating repulsive interactions, which “push” the two ligands closer together and encourage bond formation between them. This should also be appropriately cited. In this case, the -N(SiMe₃)₂ substituent gets eliminated upon the reaction of the NHC-Sn(II) complex with a C-H bond. In my opinion calling it “destabilized” is incorrect – it implies that it becomes less stable upon complexation. But it does not decompose. On the contrary, it forms macrocycles. The NHC adds to the Sn center making an energetically more favored species that has more reactive Sn center. So basically, NHC activates the Sn complex. Therefore, I would completely refrain from using the term “destabilized” in the title or anywhere in the text. There are many interesting points in the article and in my opinion.
3. “This can be attributed to the decreased ability of the heavier tetrel elements to form sp-hybrid orbitals, which consequently prevents the formation of stable NHC–tin adducts.[15]”
Is this sentence correct? Why does the decreased ability of the heavier tetrel elements to form sp-hybrid prevent the formation of stable NHC–tin adducts? If the adduct formation is via coordination to a vacant p orbital of the stannylene, shouldn't the decreased ability to hybridization facilitate the adduct formation?
4. “Furthermore, the parent tetrylenes E(14)H₂ exhibit decreased Lewis acidity and Lewis basicity when descending Group 14, making the formation of stable NHC– stannylene complexes more difficult.[16]”
How does decreased Lewis basicity make the formation of stable NHC– stannylene complexes more difficult?
5. The addition of a Lewis base to the Sn center facilitates its ability to react with the C-H bond and eliminate the HN(SiMe₃)₂ substituent. It should be clarified what happens to the Sn center upon coordination: increased Lewis basicity? Increased steric bulk that needs to be released? What is the driving force of the reaction?
6. Why out of all the possible reaction sites the Sn center of 2a preferentially react with the C-H bonds of the methyl of an additional molecule of 2a. Why, for example, not the C-H bonds on the -N(SiMe₃)₂ substituent, or the C-H bonds of the HN(SiMe₃)₂ by-product? The authors should comment on that. Also, did the authors try to trap this adduct by other

substrates? This is related to the previous comment.

7. By how much is the insertion of the Sn-NHC adduct into the C-H bond kinetically preferred over the insertion of the Sn complex without the NHC into the C-H? The addition of NHC to the Sn center makes the species reactive – this appears to be the main point of the article. Yet, the comparison of the “naked” Sn with the Sn-NHC adduct is missing. Pretty much only the NPA charges are brought for comparison.

8. The mechanism proposed in Fig.3. The authors propose a concerted, metathesis-type mechanism in which the N-H and Sn-C bonds are formed while the Sn-N and bonds are broken. Simultaneously, the $\text{HN}(\text{SiMe}_3)_2$ by-product is released. The authors should mention what happens if this proceeds via a two-step process – oxidative addition into the C-H bond, forming the hypercoordinate Sn(IV), followed by reductive elimination. Penta-coordinate Sn compounds are not uncommon and have been isolated. They can certainly form reactive intermediates. If the mechanism is indeed concerted, then there is an interesting point of metal-ligand cooperativity in C-H activation. It should be emphasized and elaborated. But even if the two-step mechanism will prove to be kinetically more favorable, the fact that the metal-ligand cooperative mechanism has a relatively low barrier is already interesting for future investigations.

9. “sufficiently clean NMR spectra for the characterization of both 3a and 4b,c in solution could not be obtained due to their poor solubility in various organic solvents (aliphatic and aromatic hydrocarbons, ethers, etc.), especially when the crystalline materials were dried under vacuum.”

What about solid-state NMR?

10. The electronic structure description of compound 2 should be more detailed and compared to 1. Especially, features that would differentiate their nucleophilicity and reactivity.

11. “Natural Population Analysis (NPA) found a significant decrease in NPA charge (1.29 to 1.08) at the tin centre, showing higher electron density at the metal. This is reflected in the Wiberg bond index for the Sn–N interaction, which decreases from 0.45 to 0.38, indicating a reduction in bond strength and thus allowing for the more facile activation across the bond...” I don’t understand how these two sentences are connected.

12. “Having established a rationale for the impact of the different terphenyl ligands on the reactivity of the respective heteroleptic stannylenes”...

What is the rationale - the steric demand? I don’t see how it was established. Simply stating that it is so does not make a compelling case.

13. “Abstraction of the proton then proceeds with a ΔG^\ddagger_{298} of +22.5 kcal•mol⁻¹, in line with the requirement to heat the reaction, producing the amine and heterocycle 5a with an ΔG_{298} of -10.9 kcal•mol⁻¹ .”

Why is this pathway preferred over the C-H bond activation of the Me? By how much?

According to the Eyring equation, the half-life of a first-order reaction in this case, A to 5a, with ΔG^\ddagger of 28.1 kcal mol⁻¹ at 60 °C, is 75.3 hours. It should take the reaction almost 19 days to reach 98% conversion in these conditions. The reported value it is not really “inline”, but is much higher than what is observed experimentally. Is it really a concerted mechanism? Can it be similar to the C-H activation by other known terphenyls, where the C-H activation proceeds in two steps – first, a proton abstraction and then a reaction with a C=C double bond? In the case of the authors, this should form the corresponding tin-hydride. Then, a reductive elimination should form the resulting products.

14. Related to the previous comment, BP86 is not a good method to calculate reaction barriers. BP86 is notoriously prone to self-interaction error and basis set superposition error. I would suggest using the obtained geometries, which should be good at PB86-D4/QZ level (but should still be justified), and carrying out single-point calculations with more appropriate methods. This also applies to other reactions reported in the article. Also, please calculate and comment on the viability of the alternative mechanism for the A to 5a transformation, i.e., a two-step insertion of the Sn center into the C-H bond, followed by the reductive elimination.

15. In all cases, carbenes with a methylated backbone were used, which allows for the C-H activation. The authors should say what happens when this reaction pathway is blocked by reacting the stannylene with a small carbene that has hydrogens instead of the methyl substituents.

16. The mechanism for the formation of 6 should at least be proposed. It would be better if it were calculated.

Reviewer #2

(Remarks to the Author)

From the perspective of reaction novelty, Malte Fischer and colleagues examine the influence of N-heterocyclic carbene (NHC) ligation on heteroleptic stannylenes, highlighting enhanced reactivity patterns. This ligation leads to formal intermolecular and intramolecular C(sp³)-H activations at the methyl groups within the backbone of selected NHCs, producing a diverse range of products. However, compounds 2a and 2b have previously demonstrated reactivity of the Sn–N moiety towards C(sp³)-H activations, which somewhat reduces the novelty of this particular aspect of the current study. Regarding the products, the research shows that these reactions are highly dependent on the adjacent steric hindrance

provided by both the chosen NHC and the terphenyl ligands surrounding the tin center, ultimately facilitating the selective synthesis of novel stannylene heterocyclic complexes. Nonetheless, concerning tin-based macrocyclic molecules, several examples have already been reported in the literature. The tin heterocyclic compounds mentioned in this work may exhibit no reactivity and have no application as macrocyclic molecules, potentially due to solubility issues. It would be more meaningful to further explore the reactivity of compounds 2a-2c, perhaps through catalysis. Therefore, based on the current evaluation, publication in Nature Communications is not recommended. However, I have a few comments and suggestions for the authors to consider:

1. The manuscript describes the synthesis of numerous stannylene metallocycle compounds, yet their reactivity is not demonstrated. Could these compounds activate small molecules, such as ammonia?
2. The main text states that heating compound 1a at 80°C for 3 hours can completely remove HN(SiMe₃)₂. However, the NMR data in the supporting literature corresponds to the heating process that removes HN(SiMe₃)₂ from compound 2a. Please verify whether this is a typographical or writing error.
3. The compounds 3a, 4b, and 4c lack corresponding NMR data, and all result from respective in-situ reactions. Is their isolation yield reproducible?
4. Why are there two signal peaks in the ¹¹⁹Sn NMR spectrum of compound 2b?
5. The main text states that compound 2a decomposes into non-separable compounds in solution but does not mention the stability of compounds 2b and 2c in solution. Exploring the reactivity of compounds 2a-2c further, such as through catalysis, would be more meaningful.

Reviewer #3

(Remarks to the Author)

The manuscript by Fischer and Townrow reports a set of aminoterphenylstannylenes and their peculiar reactivity towards NHCs. The latter contain methyl groups at the backbone which for the IMe₄ undergo CH-activation (deprotonation) leading to NHC bridged cyclic oligomers with each stannylene connected to a carbene center and an anionic methylide carbon atom. The terphenyl substituent influences the ring size of the resulting tin-containing macrocycles. Structural characterization includes the initial metastable regular adduct and the resulting dimeric and tetrameric macrocycles. A mechanistic proposal is backed by DFT calculations in a convincing manner. This approach is straightforward and consistent using IMe₄ whereas the carbene liPr₂Me₂ behaves differently. For the 2,6-dimesphenylsubstituted stannylene, a mesityl methyl group is deprotonated leading to an all carbon substituted stannylene adduct. For the larger terphenyls, the terphenyl substituent is partially cleaved off along with partial deprotonation of the hexamethyldisilazanide and Sn-Sn bond formation. Overall, this article reports remarkable findings and is well written. There are only few points of improvement as outlined below.

The authors should come up with a reasonable suggestion explaining the formation of the Sn-Sn bonded species.

I'd suggest adding the eliminated species HNTMS₂ and TerH directly into the schemes so that they are more intuitive to follow.

Figure 1: ... "with Select Nuclearity"??

First paragraph of Results and discussion: skip "after the measurement of the heteronuclear and 2D NMR spectra"

In discussing the Sn-C(NHC) bond length this is categorized as elongated (2.3255(9) Å) which is explained by the donor properties of the adjacent amido substituent. However, several stannylene NHC adducts lacking amido or pronounced π-donor substituents have comparable or even longer Sn-C(NHC) bond lengths, e.g. Cl₂Sn-NHC (2.341(8) Å) or >P₂Sn-NHC (2.313(2) Å). The latter two stannylene adducts are also missing in the row of "additional examples within this compound class" on the first page (Chem. Commun., 2009, 7119; Chem. - Eur. J., 2018, 24, 16774).

Version 1:

Reviewer comments:

Reviewer #1

(Remarks to the Author)

The authors revised the manuscript following the comments by reviewers.

A response letter contains the point-by-point answers by the authors and most of the points are clarified.

However, I still see some issues with the current version.

Concerning the bonding situation and electronic nature of 2 (points 5, 6, and 10), the authors responded to describe what they are assuming and added the molecular orbitals discussion.

However, the comparison of 2 with the starting material 1 is the key point of this manuscript.

The authors stated the addition of NHC provides a more electron-rich tin center and weakens the Sn-N bond.

These points should be discussed with the computed results in 1 and 2 (nucleophilicity/basicity at the Sn center, NBO analysis, bond order for the Sn-N bond, etc.).

Another concern is the reaction mechanism for the formation of 6 (point 16).

This reaction looks a bit different pathway from the other presented reactions in the paper, and the product contains the formation of the Sn-Sn bond.

Since the other presented reactions were well studied by DFT calculations, the authors should investigate this reaction mechanism by computationally, which will certainly improve the quality of the manuscript.

Reviewer #2

(Remarks to the Author)

In this resubmitted manuscript, the authors did significant changes, to address the critical comments from both reviewers on the previous version. Therefore, acceptance is recommended.

Reviewer #3

(Remarks to the Author)

In the revised version of the manuscript by Fischer and Townrow all issues have been addressed in full. I appreciate the modified title avoiding the term "destabilized". I am convinced this is a very nice contribution to Nature Communications and do recommend acceptance of the manuscript.

Version 2:

Reviewer comments:

Reviewer #1

(Remarks to the Author)

I find the reaction mechanism for the formation of 6b,c are very interesting, since it differs from the other reactions in this paper presented. I still believe that it is important to at least provide some hints or the initial key step derived from computational methods.

However, it seems that the authors do not wish to pursue further elucidation of this reaction mechanism in this paper, so I leave the final decision on this matter to the editor.

Reviewer 1	
Carbene-destabilised stannylenes: Accessing selective C(sp³)–H bond scission at the steric limit” article by Townrow, Fischer, et al. reports on the inter- and intramolecular activation of C-H bonds by a low oxidation state Sn center. The activation is induced upon the reaction of Sn(II) with NHCs. Overall, the work is of interest to organometallic and main-group chemists. Below are several points that the authors should address before the publication:	We sincerely thank the reviewer for the detailed and thoughtful feedback. We have carefully considered each point raised and addressed them as outlined below:
1. The principle reported here is conceptually similar to that reported already. The addition of a Lewis base facilitates the reactivity of the Sn(II) center that is accompanied by the release of HN(SiMe₃)₂ (Ref. 43,44). The authors should make the point of novelty more clearly.	Thank you for the comment. We think that the part in the introduction emphasizing that NHC activation has not been observed in any cases when ligated to stannylenes underlines the novelty described herein. We further added a sentence that our previous work describes formal C(sp³)–H activation of a substrate which is known to be deprotonated to give the corresponding ylides whereas the herein depicted type of NHC activation has not been observed before. We further included a paragraph drawing parallels to mesoionic olefins (mNHOs), as the activation of the backbone methyl group in the NHC–stannylyne adducts allows the tin-bridging former NHC units to be regarded as hybrid NHC/mNHO ligands in the macrocycles.
2. One of my biggest concerns is about the presentation. I understand the authors try to make a catchy title, however in my opinion it really hurts the chemistry carried out here. Referring to the NHC-stannylyne complexes as “carbene-destabilized” is very odd. I completely disagree with this type of statement. The complexation may make the Sn center more nucleophilic, and more reactive; it may induce reactivity of which the “naked” Sn complex is not capable. The introduction of bulkier substituent makes the Sn center more prone to reductive elimination – this is a known phenomenon. Steric effects from bulky ligands can increase the rate of reductive elimination by creating repulsive interactions, which “push” the two ligands closer together and encourage bond formation between them. This should also be appropriately cited. In this case, the -N(SiMe₃)₂ substituent gets eliminated upon the reaction of the NHC-Sn(II) complex with a C-H bond. In my opinion calling it “destabilized” is incorrect – it implies that it becomes less stable upon complexation. But it does not decompose. On the contrary, it forms macrocycles. The NHC adds to the Sn center making an energetically more favored species that has more reactive Sn center. So basically, NHC activates the Sn complex. Therefore, I would completely refrain from using the term “destabilized” in the title or anywhere in the text. There are many interesting points in the article and in my opinion.	After reconsidering the chosen title, our implication on calling these complexes “carbene-destabilized” was to emphasize the special position of these NHC-stannylyne adducts in comparison to all other hitherto reported ones and after the comment, we can definitely agree why this can cause confusion or even be called “wrong”. We have rephrased the respective passages in the manuscript. Regarding the effect of sterics this has now been added to the manuscript. “NHC ligation can also be understood as a driving force for steric overcrowding at the tin centre, promoting the favourable release of HN(SiMe₃)₂. Similar observations have been noted, for example, in the reactions of homoleptic bis(terphenyl)-substituted tetrylenes, (MesTer)₂Ge and (DippTer)₂Ge, with dihydrogen. The less bulky derivative forms the corresponding dihydride via oxidative addition, whereas the sterically more demanding system produces the trihydride DippTerGeH₃ along with DippTerH as a side product.” Reference: Peng, Y., Guo, J.-D., Ellis, B. D., Zhu, Z., Fettinger, J. C., Nagase, S. & Power, P. P. Reaction of Hydrogen or Ammonia with Unsaturated Germanium or Tin Molecules under Ambient Conditions: Oxidative Addition versus Arene Elimination. J. Am. Chem. Soc. 131, 16272-16282 (2009). We also changed the title to: “Carbene-activated stannylenes: Accessing selective C(sp³)–H bond scission at the steric limit”

3. "This can be attributed to the decreased ability of the heavier tetrel elements to form sp-hybrid orbitals, which consequently prevents the formation of stable NHC–tin adducts.[15]" Is this sentence correct? Why does the decreased ability of the heavier tetrel elements to form sp-hybrid prevent the formation of stable NHC–tin adducts? If the adduct formation is via coordination to a vacant p orbital of the stannylene, shouldn't the decreased ability to hybridization facilitate the adduct formation?	Thanks for pointing that out. We changed the paragraph accordingly.
4. "Furthermore, the parent tetrylenes E(14)H2 exhibit decreased Lewis acidity and Lewis basicity when descending Group 14, making the formation of stable NHC– stannylene complexes more difficult.[16]" How does decreased Lewis basicity make the formation of stable NHC– stannylene complexes more difficult?	This has been corrected.
5. The addition of a Lewis base to the Sn center facilitates its ability to react with the C-H bond and eliminate the HN(SiMe3)2 substituent. It should be clarified what happens to the Sn center upon coordination: increased Lewis basicity? Increased steric bulk that needs to be released? What is the driving force of the reaction? 6. Why out of all the possible reaction sites the Sn center of 2a preferentially react with the C-H bonds of the methyl of an additional molecule of 2a. Why, for example, not the C-H bonds on the -N(SiMe3)2 substituent, or the C-H bonds of the HN(SiMe3)2 by-product? The authors should comment on that. Also, did the authors try to trap this adduct by other substrates? This is related to the previous comment.	We have added the following sentence to the manuscript to answer this with our working hypothesis: ..., we sought to investigate how the introduction of strong Lewis bases like NHCs might impact this observed activation pattern and ultimately if this might trigger NHC activation. We hypothesized that NHC-ligation renders the tin centre more electron rich to further weaken the tin–nitrogen bond, whilst sterically repelling the trimethylsilyl groups and thus rendering these complexes more reactive towards C–H activation with the leaving group HN(SiMe₃)₂ and accordingly increasing the reactivity across the tin–nitrogen bond.
7. By how much is the insertion of the Sn-NHC adduct into the C-H bond kinetically preferred over the insertion of the Sn complex without the NHC into the C-H? The addition of NHC to the Sn center makes the species reactive – this appears to be the main point of the article. Yet, the comparison of the "naked" Sn with the Sn-NHC adduct is missing. Pretty much only the NPA charges are brought for comparison.	We have computationally modelled a theoretical reaction between 1a and 2a, finding that the process is much less favourable ($\Delta G^{\ddagger}_{298} = 37.4$ kcal•mol⁻¹). We thank the reviewer for helping make our study more thorough.
8. The mechanism proposed in Fig.3. The authors propose a concerted, metathesis-type mechanism in which the N-H and Sn-C bonds are formed while the Sn-N and bonds are broken. Simultaneously, the HN(SiMe3)2 by-product is released. The authors should mention what happens if this proceeds via a two-step process – oxidative addition into the C-H bond, forming the hypercoordinate Sn(IV), followed by reductive elimination. Penta-coordinate Sn compounds are not uncommon and have been isolated. They can certainly form reactive intermediates. If the mechanism is indeed concerted, then there is an interesting point of metal-ligand cooperativity in C-H activation. It should be emphasized and elaborated. But even if the two-step mechanism will prove to be kinetically more favorable, the fact	We have investigated an oxidative addition route computationally. For the intramolecular system, this was found to be unfavourable and proceeds via a much higher energy barrier (+41 kcal/mol above the starting materials, producing the Sn(IV) hydride intermediate at +19 kcal/mol). We thank the reviewer for their interest and have added these results to the supporting information. For the intermolecular mechanism, an intermediate with comparable energy was found, however steric hinderance prevented relaxed surface scans towards the starting materials and products, meaning that the transition states could

that the metal-ligand cooperative mechanism has a relatively low barrier is already interesting for future investigations. 9. "sufficiently clean NMR spectra for the characterization of both 3a and 4b,c in solution could not be obtained due to their poor solubility in various organic solvents (aliphatic and aromatic hydrocarbons, ethers, etc.), especially when the crystalline materials were dried under vacuum." What about solid-state NMR?	not be located. It would be reasonable to assume that high steric hinderance around the Sn centres presented in this manuscript promote the cooperative pathway over the oxidative addition path.
10. The electronic structure description of compound 2 should be more detailed and compared to 1. Especially, features that would differentiate their nucleophilicity and reactivity.	We have included a more detailed description of the molecular orbitals comparing 2 to 1 in order to rationalise the observed reactivity.
11. "Natural Population Analysis (NPA) found a significant decrease in NPA charge (1.29 to 1.08) at the tin centre, showing higher electron density at the metal. This is reflected in the Wiberg bond index for the Sn–N interaction, which decreases from 0.45 to 0.38, indicating a reduction in bond strength and thus allowing for the more facile activation across the bond..." I don't understand how these two sentences are connected.	By increasing electron density at the tin centre, and lengthening the Sn–N interaction, the carbene increases the reactivity of the Sn–N bond, resulting in the release of HMDS.
12. "Having established a rationale for the impact of the different terphenyl ligands on the reactivity of the respective heteroleptic stannylenes"... What is the rationale - the steric demand? I don't see how it was established. Simply stating that it is so does not make a compelling case.	This has been adjusted.
13. "Abstraction of the proton then proceeds with a ΔG^\ddagger 298 of +22.5 kcal·mol⁻¹, in line with the requirement to heat the reaction, producing the amine and heterocycle 5a with an ΔG_{298} of -10.9 kcal·mol⁻¹." Why is this pathway preferred over the C-H bond activation of the Me? By how much? According to the Eyring equation, the half-life of a first-order reaction in this case, A to 5a, with ΔG^\ddagger of 28.1 kcal mol⁻¹ at 60 °C, is 75.3 hours. It should take the reaction almost 19 days to reach 98% conversion in these conditions. The reported value it is not really "inline", but is much higher than what is observed experimentally. Is it really a concerted mechanism? Can it be similar to the C-H activation by other known tetrylenes, where the C-H activation proceeds in two steps – first, a proton abstraction and then a reaction with a C=C double bond? In the case of the authors, this should form the corresponding tin-hydride. Then, a reductive elimination should form the resulting products.	We thank the reviewer for their concern; however, we disagree with their point of view. Many reactions with similar DFT-calculated barriers proceed either slowly at room temperature, or with some heating and is not "much higher than what is observed experimentally". We have looked into the alternative mechanism, and found it to be thermodynamically unviable.
14. Related to the previous comment, BP86 is not a good method to calculate reaction barriers. BP86 is notoriously prone to self-interaction error and basis set superposition error. I would suggest using the obtained geometries, which should be good at PB86-D4/QZ level (but should still be justified), and carrying out single-point calculations with more appropriate methods. This also applies to other reactions reported in the	We have found that this level of theory has been more in line with the experimental findings for our systems than other methods, for example with the use of hybrid functionals (e.g. PBE0-D3BJ/def2-TZVP). There are also many literature examples of this level of theory agreeing with experimental findings. For this, we would draw the reviewer's

article. Also, please calculate and comment on the viability of the alternative mechanism for the A to 5a transformation, i.e., a two-step insertion of the Sn center into the C-H bond, followed by the reductive elimination.	attention to the Structure and Bonding volume 167 "Computational Studies in Organometallic Chemistry" where many examples are given, in particular those showing that BP86-D3BJ has in the past predicted transition states within 1% of the experimentally measured barrier. The use of quadruple zeta would be, in the opinion of the authors, an unnecessary use of computational resources with little gain.
15. In all cases, carbenes with a methylated backbone were used, which allows for the C-H activation. The authors should say what happens when this reaction pathway is blocked by reacting the stannylene with a small carbene that has hydrogens instead of the methyl substituents.	We exemplarily investigated the reactivity of compounds 1 toward freshly synthesized IPr. Upon heating the reaction mixtures to first 80 °C and then 100 °C only decomposition of the tin starting materials was observed without consumption of the respective NHC.
16. The mechanism for the formation of 6 should at least be proposed. It would be better if it were calculated.	Since the reaction is rather abstract, and we only have this data point, we feel that it would be unscientific to propose a mechanism here. We are currently following this result up for a future study and plan to be able to investigate the mechanism then.
Reviewer 2	
From the perspective of reaction novelty, Malte Fischer and colleagues examine the influence of N-heterocyclic carbene (NHC) ligation on heteroleptic stannylenes, highlighting enhanced reactivity patterns. This ligation leads to formal intermolecular and intramolecular C(sp³)-H activations at the methyl groups within the backbone of selected NHCs, producing a diverse range of products. However, compounds 2a and 2b have previously demonstrated reactivity of the Sn-N moiety towards C(sp³)-H activations, which somewhat reduces the novelty of this particular aspect of the current study. Regarding the products, the research shows that these reactions are highly dependent on the adjacent steric hindrance provided by both the chosen NHC and the terphenyl ligands surrounding the tin center, ultimately facilitating the selective synthesis of novel stannylene heterocyclic complexes. Nonetheless, concerning tin-based macrocyclic molecules, several examples have already been reported in the literature. The tin heterocyclic compounds mentioned in this work may exhibit no reactivity and have no application as macrocyclic molecules, potentially due to solubility issues. It would be more meaningful to further explore the reactivity of compounds 2a-2c, perhaps through catalysis. Therefore, based on the current evaluation, publication in Nature Communications is not recommended. However, I have a few comments and suggestions for the authors to consider: 1. The manuscript describes the synthesis of numerous stannylene metallocycle compounds, yet their reactivity is not demonstrated. Could these compounds activate small molecules, such as ammonia?	We sincerely thank the reviewer for their time and effort in providing a thoughtful and detailed evaluation of our manuscript. The primary focus of our work is indeed on the synthesis and characterization of what we believe to be a unique extension of NHC reactivity, enabling the isolation of well-defined macrocycles. These outcomes are highly dependent on both the nature of the supporting ligand framework at tin and the choice of NHC, rather than being centred on catalytic applications. While we fully intend to explore the reactivity of these complexes in future studies, this lies beyond the scope of the current communication.

2. The main text states that heating compound 1a at 80°C for 3 hours can completely remove HN(SiMe ₃) ₂ . However, the NMR data in the supporting literature corresponds to the heating process that removes HN(SiMe ₃) ₂ from compound 2a. Please verify whether this is a typographical or writing error.	This was a writing error and has been corrected accordingly.
3. The compounds 3a, 4b, and 4c lack corresponding NMR data, and all result from respective in-situ reactions. Is their isolation yield reproducible?	Yes. We have included the monitoring of the reaction progress for the formation of the complexes 3a , 4b and 4c , as well as images for illustration, in the Supporting Information (SI). During the course of reproducing the material, we also obtained crystalline material of 4b without co-crystallising solvent. This has also been added to the revised version of the manuscript.
4. Why are there two signal peaks in the 119Sn NMR spectrum of compound 2b?	The second signal belongs to an unknown impurity, accounts for less than 3% as indicated by integration of the respective signals, and is now noted in the figure caption.
5. The main text states that compound 2a decomposes into non-separable compounds in solution but does not mention the stability of compounds 2b and 2c in solution. Exploring the reactivity of compounds 2a-2c further, such as through catalysis, would be more meaningful.	The statement regarding the decomposition of compound 2a in solution is a quote from reference [50], a manuscript published during the final stages of our work. Herein, we present what was referred to as decomposition in that reference as instead a follow-up reactivity. The non-stability of 2b and 2c in solution is definitely demonstrated in our work by their conversion to macrocycles 4b and 4c , respectively. In said reference, Inoue and co-workers indeed report on the application of these compounds in hydrosilylation catalysis of carbonyl compounds.
Reviewer 3	
The manuscript by Fischer and Townrow reports a set of aminoterphenylstannylenes and their peculiar reactivity towards NHCs. The latter contain methyl groups at the backbone which for the IMe ₄ undergo CH-activation (deprotonation) leading to NHC bridged cyclic oligomers with each stannylene connected to a carbene center and an anionic methyllide carbon atom. The terphenyl substituent influences the ring size of the resulting tin-containing macrocycles. Structural characterization includes the initial metastable regular adduct and the resulting dimeric and tetrameric macrocycles. A mechanistic proposal is backed by DFT calculations in a convincing manner. This approach is straightforward and consistent using IMe ₄ whereas the carbene liPr ₂ Me ₂ behaves differently. For the 2,6-dimesphenylsubstituted stannylene, a mesityl methyl group is deprotonated leading to an all carbon substituted stannylene adduct. For the larger terphenyls, the terphenyl substituent is partially cleaved off along with partial deprotonation of the hexamethyldisilazanide and Sn-Sn bond formation. Overall, this article reports remarkable findings and is well written. There are only few points of improvement as outlined below.	We sincerely thank the reviewer for their thoughtful and positive evaluation of our manuscript. We are grateful for your suggestions for improvement, which we carefully addressed to further enhance the quality and clarity of the manuscript.

The authors should come up with a reasonable suggestion explaining the formation of the Sn-Sn bonded species.	Whilst we do not know the mechanism, aryl-substituted tin complexes are known, on loss of their ligands, to form Sn-Sn bonded species, ultimately forming clusters as thermodynamic sinks. See 10.1039/C4SC00365A. We thank the reviewer for their interest and this has been added to the manuscript to provide clarity.
I'd suggest adding the eliminated species HNTMS2 and TerH directly into the schemes so that they are more intuitive to follow.	We have included the by-products in the figures.
Figure 1: ... "with Select Nuclearity"??	The sentence has been rephrased.
First paragraph of Results and discussion: skip "after the measurement of the heteronuclear and 2D NMR spectra"	Done.
In discussing the Sn-C(NHC) bond length this is categorized as elongated (2.3255(9) Å) which is explained by the donor properties of the adjacent amido substituent. However, several stannylene NHC adducts lacking amido or pronounced π-donor substituents have comparable or even longer Sn-C(NHC) bond lengths, e.g. Cl₂Sn-NHC (2.341(8) Å) or >P₂Sn-NHC (2.313(2) Å). The latter two stannylene adducts are also missing in the row of "additional examples within this compound class" on the first page (Chem. Commun., 2009, 7119; Chem. - Eur. J., 2018, 24, 16774).	We thank the reviewer for pointing that out. We changed the discussion accordingly and added the missing examples to the references.

Reviewer 1	
The authors revised the manuscript following the comments by reviewers. A response letter contains the point-by-point answers by the authors and most of the points are clarified. However, I still see some issues with the current version.	We sincerely thank the reviewer once again for their invaluable feedback, which has greatly contributed to improving and clarifying the presented study. Below, we address the two remaining issues as follows:
Concerning the bonding situation and electronic nature of 2 (points 5, 6, and 10), the authors responded to describe what they are assuming and added the molecular orbitals discussion. However, the comparison of 2 with the starting material 1 is the key point of this manuscript. The authors stated the addition of NHC provides a more electron-rich tin center and weakens the Sn-N bond. These points should be discussed with the computed results in 1 and 2 (nucleophilicity/basicity at the Sn center, NBO analysis, bond order for the Sn-N bond, etc.).	We have to apologize for not highlighting this section in the “track changes version” of the manuscript. We would like to draw the reviewer’s attention to the following discussion, which is now marked in the revised manuscript within the updated “track change version:” “We have previously demonstrated the reactivity of the Sn–N moiety in 1 towards C–H bonds, resulting in the production of bis(trimethylsilyl)amine and a tin alkyl complex, albeit with mildly activated C–H bonds.^[45,46] Compounds 2, on the other hand, are able to activate aryl-substituted methyl groups, which are typically more difficult to activate. Natural Population Analysis (NPA) found a significant decrease in NPA charge (1.29 to 1.08) at the tin centre, showing higher electron density at the metal (i.e. increased basicity). This is reflected in the Wiberg bond index for the Sn–N interaction, which decreases from 0.45 to 0.38, indicating a reduction in bond strength and thus allowing for the more facile activation across the bond; and supports the observed elongation of the bond measured by SCXRD, vide supra. We have additionally added the relation to basicity to guide the reader.
Another concern is the reaction mechanism for the formation of 6 (point 16). This reaction looks a bit different pathway from the other presented reactions in the paper, and the product contains the formation of the Sn-Sn bond. Since the other presented reactions were well studied by DFT calculations, the authors should investigate this reaction mechanism by computationally, which will certainly improve the quality of the manuscript.	For the other reactions in the paper, the mechanisms seemed rather intuitive since there was a formal C-H activation and common by-product (HN(SiMe₃)₂). These follow the narrative of this manuscript. The formation of 6, on the other hand, is more complex and results in something very different to the other reactions. Due to the outcome of the reaction and formal disproportionation at tin with regard to the starting material, an open-shell pathway seems likely involving tin alkyl species which are known to be prone to radical processes. It would be definitely interesting to know the mechanism behind its formation, but with one datapoint, it would be entirely speculative. Added to the robustness of study in the other reactions, we feel that with our current level of understanding, this would negatively impact on the quality of the manuscript. These findings serve as a preview to ongoing work in our groups, which we decided that the reader would appreciate (as opposed to simply leaving this reaction out of the manuscript as the result does not fit with the other reactions).

	We hope that the reviewer can see our point of view on these matters.
Reviewer 2	
In this resubmitted manuscript, the authors did significant changes, to address the critical comments from both reviewers on the previous version. Therefore, acceptance is recommended.	We sincerely thank the reviewer for their time and dedication, and we deeply appreciate their recommendation for accepting our study following the revisions.
Reviewer 3	
In the revised version of the manuscript by Fischer and Townrow all issues have been addressed in full. I appreciate the modified title avoiding the term "destabilized". I am convinced this is a very nice contribution to Nature Communications and do recommend acceptance of the manuscript.	We sincerely thank the reviewer for their positive evaluation of our manuscript and their valuable, constructive feedback, which has been instrumental in improving our study.